# Advances of the Cubic Symmetry Crystalline Systems to Create Complex, Bright Luminescent Ceramics

**Valentina Smyslova** [1], **Daria Kuznetsova** [1], **Alexey Bondaray** [2], **Petr Karpyuk** [1], **Mikhail Korzhik** [1,2,*],
**Ilya Komendo** [1], **Vladimir Pustovarov** [3], **Vasilii Retivov** [1] and **Dmitry Tavrunov** [3]

[1]  National Research Center "Kurchatov Institute", 123098 Moscow, Russia; smyslovavg@gmail.com (V.S.);
daria_kyznecova@inbox.ru (D.K.); silancedie@mail.ru (P.K.); i.comendo@gmail.com (I.K.);
vasilii_retivov@mail.ru (V.R.)
[2]  Institute for Nuclear Problems, Belarus State University, 11 Bobruiskaya, 220030 Minsk, Belarus;
alesonep@gmail.com
[3]  Experimental Physics Department, Ural Federal University, 620075 Yekaterinburg, Russia;
vpustovarov@bk.ru (V.P.); mr.tavrunov@mail.ru (D.T.)
*   Correspondence: mikhail.korjik@cern.ch

**Abstract:** A method to create compositionally disordered compounds with a high number of cations in the matrices, that utilize the cubic spatial symmetry of the garnet-type crystalline systems is demonstrated. Mixtures of the garnet-type powdered materials solely doped with Ce were used to create atomic compositions of high complexity. Several mixed systems, namely $Gd_3Al_2Ga_3O_{12}/(Gd,Y)_3Al_2Ga_3O_{12}$, $Y_3Al_5O_{12}/Gd_3Al_2Ga_3O_{12}$, and $Y_3Al_5O_{12}/Y_3Al_2Ga_3O_{12}$ were annealed, compacted and sintered in air. The materials were evaluated for structural, luminescence, and scintillation properties. It was demonstrated that the properties of the resulting ceramics are a little dependent on the granularity of powders when the median particle size is below ~5 μm.

**Keywords:** compositional disorder; scintillator; ceramics; garnet; cerium; luminescence; scintillation yield

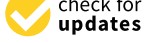



## 1. Introduction

Compositionally disordered crystalline materials with a garnet structure form a unique platform for creating luminescent materials for lighting and detecting various types of ionizing radiation [1–12]. Similarly to luminescent inorganic heterostructores [13–15], they form a list of the advanced materials for photonics applications. Simultaneously, the creation of increasingly disordered systems necessitates well-controlled methods for the preparation of intermediate products (powders), which are then compacted and annealed to obtain dense translucent or transparent ceramics [16]. The production of transparent and translucent ceramics based on garnet compounds is a complex technological process that includes three major stages: the synthesis of a precursor from the specified chemicals, the formation of a "green body", and high-temperature sintering to obtain a dense polycrystalline material. The crucially important step is the synthesis of the precursor, since the precise control of the composition and specific microstructure of particles affects the functional characteristics and microstructure of the resulting ceramics. The difficulties in precise control of synthesis conditions when the number of cations in the composition is increased also make the synthesis stage more complex [17–19]. Variations of the co-precipitation method for producing the powders of the garnet compounds have proven themselves as quite useful [20–23]. At the same time, intentions to move from binary and ternary compounds to more complex quaternary and quintuple compositions, containing a larger number of cations, inevitably lead to problems with the precise control of the composition of the compound due to effects associated with the different solubilities of rare earth (RE) elements in the compound. A violation, for example, of the stoichiometry of the compound can lead to the leveling of

the advantages introduced into the compound by compositional disorder in the cationic subsystem [24,25].

It is worth noting that the combination of the cubic spatial symmetry and the similarity of the lattice parameters of aluminum-gallium garnet crystalline systems, based on the RE elements and yttrium, opens the door for new technological operations in the creation of increasingly complex ceramics. One promising approach to the preparation of the complex compounds could be the mixing of compounds from the same structural series with a lower number of cations in the composition for further sintering of high-complexity ceramics. For example, it can be expected that mixing the binary $Y_3Al_5O_{12}$ (YAG) and the ternary $Gd_3(Al,Ga)_5O_{12}$ (GAGG) compounds in a certain proportion, of which the powder technology process is well developed, will result in the quaternary compound $(Gd,Y)_3(Al,Ga)_5O_{12}$ (GYAGG), which, when doped with Ce ions, has attractive scintillation properties [26]. Note that this approach makes it possible to achieve two extreme cases: the formation of a disordered compound with a uniform random distribution of cation atoms in the crystal lattice, or a set of adjacent crystallites of different cationic compositions or even with different activator ions. Obviously, the intermediate case, in which individual crystallites are distributed in a lattice with mixed atoms, is also of practical interest. At the same time, the implementation of one or another case significantly depends on the granulometric composition of powders of simpler compounds used to obtain a mixture. It is important to keep the resulting polycationic compound within the stability of the garnet phase as well.

Motivated by the above, we investigated three mixed systems: $Gd_3(Al,Ga)_5O_{12}/(Y,Gd)_3(Al,Ga)_5O_{12}$, $Gd_3(Al,Ga)_5O_{12}/Y_3Al_5O_{12}$, and $Y_3(Al,Ga)_5O_{12}/Y_3Al_5O_{12}$. All the components, which were used for mixture preparation, were doped with Ce ions. The mixtures were used to prepare ceramic samples. The properties of the ceramics were studied by X-ray diffraction and luminescent spectroscopy, depending on the particle size distribution of the initial powders. The conclusions and discussion of the results are supported by the results of measurements of photoluminescence and scintillation properties of the obtained ceramics.

## 2. Sample Preparation and Measurements

Powder precursors of binary and ternary garnets were produced by the reverse coprecipitation method as described in [27]. All compounds were doped with Ce at a concentration x = 0.015. After calcination at 1000 °C, the powders were ground in a laboratory planetary mill (Retsch PM-100) in an isopropanol medium in various modes to obtain fractions with different average particle sizes described in Table 1. Powders were ground individually and mixed in a 1:1 ratio before sintering. Then, powders were compacted in the shape of tablets with a diameter of 20 mm by uniaxial pressing at a pressure of 64 MPa (Laboratory press PLG-20 by LabTools). The compacts were sintered in the air atmosphere at 1600 °C for 2 h using Nabertherm LHT 02/17 LB high-temperature furnace. Ceramic samples were translucent and had >99.6% theoretical density. The compositions of the samples and the corresponding abbreviations are given in Table 1 as well. Each of series 1–3 included two types of the reference samples. Reference samples 1, 2, 4, and 6 were individual compounds of the garnets obtained entirely by the coprecipitation method. Reference samples 3, 5, and 7 were also obtained by the coprecipitation method and were identical in chemical composition to those obtained by mixing powders of different granulometric compositions. Codoping of the GYAGG crystals with a small concentration of the magnesium Mg2+ (10–20 ppm) improves scintillation properties of the material, in particular the scintillation kinetics [23]. To distinguish the effects of codoping from the effects that might have been provided by applied technology, samples were not codoped with a small quantity of Mg to reduce the fraction of the slow component in scintillation. Despite this, none of the samples showed significant phosphorescence.

**Table 1.** Compositions and abbreviation of the studied samples.

| Sample Composition | Abbreviation | Median Particle Size $d_{50}$, μm | Calculated Density, g/cm³ |
|---|---|---|---|
| Serie 1 | | | |
| $Gd_{2.985}Ce_{0.015}Al_2Ga_3O_{12}$ | GAGG:Ce Ref1 * | 2 | 6.660 |
| $Gd_{1.4925}Y_{1.4925}Ce_{0.015}Al_2Ga_3O_{12}$:Ce | GYAGG:Ce Ref2 * | 2 | 5.978 |
| $Gd_{2.239}Y_{0.746}Ce_{0.015}Al_2Ga_3O_{12}$:Ce | GYAGG:Ce Ref3 | 2 | 6.294 |
| GAGG:Ce/GYAGG:Ce (1/1) | G+GY (8 μm) | 8 | 6.289 |
| GAGG:Ce/GYAGG:Ce (1/1) | G+GY (4 μm) | 4 | 6.294 |
| GAGG:Ce/GYAGG:Ce (1/1) | G+GY (2 μm) | 2 | 6.295 |
| Serie 2 | | | |
| $Y_{2.985}Ce_{0.015}Al_5O_{12}$ | YAG:Ce Ref4 | 2 | 4.552 |
| $Gd_{1.4925}Y_{1.4925}Ce_{0.015}Al_{3.5}Ga_{1.5}O_{12}$ | GYAGG Ref5 * | 2 | 5.646 |
| GAGG/YAG (1/1) | G+Y (6 μm) | 6 | 5.653 |
| GAGG/YAG (1/1) | G+Y (4 μm) | 4 | 5.655 |
| GAGG/YAG (1/1) | G+Y (2 μm) | 2 | 5.654 |
| Serie 3 | | | |
| $Y_{2.985}Ce_{0.015}Al_2Ga_3O_{12}$ | YAGG Ref6 * | 2 | 5.323 |
| $Y_{2.985}Ce_{0.015}Al_{3.5}Ga_{1.5}O_{12}$ | YAGG Ref7 * | 2 | 4.954 |
| YAGG/YAG (1/1) | YG+Y (6 μm) | 6 | 4.950 |
| YAGG/YAG (1/1) | YG+Y (3 μm) | 3 | 4.953 |
| YAGG/YAG (1/1) | YG+Y (2 μm) | 2 | 4.952 |

\* Reference compounds, which were produced from the same chemicals exclusively by the coprecipitation method.

A typical image of particles obtained after calcination is shown in Figure 1. The particles were agglomerated after calcination; the smaller crystallites have garnet habitus and are less than 1 μm in size. The particles alternate with pores, which causes a large open surface in which to agglomerate. This increases the reactivity of the milled and pressed powders at the sintering temperature.

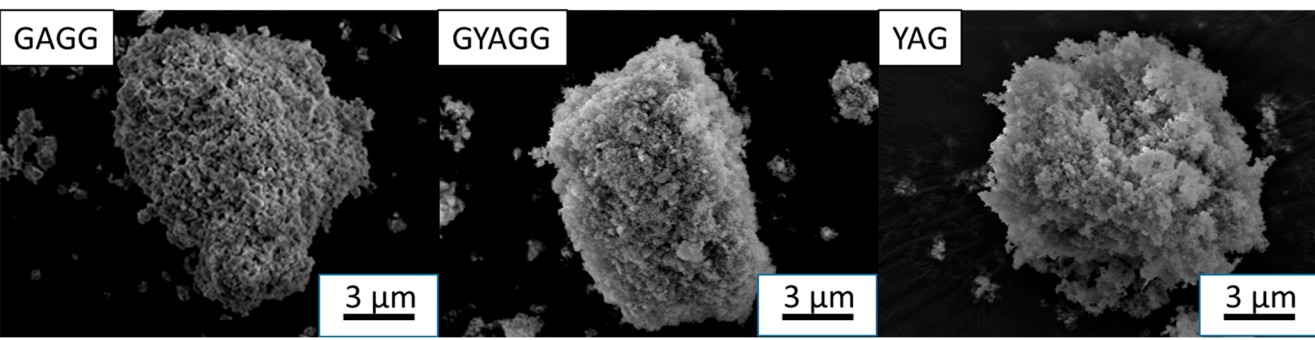

**Figure 1.** SEM images of representative powders: GAGG:Ce (Ref.1), GYAGG (Ref.2) and YAG (Ref.4) particle agglomerates after calcination at 1000 °C.

The X-ray diffraction method (2D Phaser by Bruker with linearly polarized Cu radiation kα-doublet 1.5406 and 1.5444 Å) was used to evaluate the crystal structure and lattice parameters of the samples. Data obtained were also used to evaluate the density of the material (Table 1).

The luminescence spectra at photoexcitation (PL) of the samples were measured with a Fluorat-02-PANORAMA Lumex spectrofluorimeter at room temperature in the

spectral range of 360–700 nm, with an excitation wavelength ($\lambda ex$) which correspond to the excitation in different parts of the lowest excitation band due to the $4f^1$–$5d^1f^0$-interconfiguration transition of $Ce^{3+}$ ions.

Scintillation kinetics were measured with a [22]Na source by the start-stop method with XP2020 photomultipliers in both channels, providing a response function width (FWHM) of 1.2 ns. All measurements were performed at room temperature. The relative light yield (LY) in the series was evaluated by the measurement of the full absorption peak position of [241]Am alpha-particles, as used for the measurements with translucent or powdered samples in [28]. A $Y_3Al_5O_{12}$:Ce (YAG) single crystal of 1 mm thickness, with ground surfaces and an LY of 20,600 ph/MeV(under gamma-quanta), was chosen as a reference sample. The errors in light yield measurements were defined as ±2%, whereas the fitting results for decay constants were ±2 ns, correspondingly. It was assumed that the alpha/gamma ratio of studied samples was the same.

## 3. Results

The resulting ceramic samples were single-phase with a garnet structure. Figure 2 shows the X-ray diffraction patterns of the samples from Series 1. Details of the X-ray diffraction pattern in the vicinity of the major reflex in the interval 2θ from 32 to 34 (°) are shown in Figure 3. Inset shows the calculated lattice parameters. The reflections of the compounds obtained by mixing are positioned in the interval between the mixed compounds, as expected. At the same time, the reflection for the GYAGG:Ce Ref3 sample is closer to the reflection of the GAGG:Ce Ref1 sample, since the gadolinium index in this compound X = 2.2 and exceeds that for GYAGG:Ce Ref2 (X = 1.5). The position of the peak of the major reflection measured in samples obtained from mixtures was slightly dependent on the granulometric composition of the mixed powders. The larger the median particle size, the closer the reflection of such a sample was to the reflection of the sample obtained by coprecipitation.

Figure 4 shows the X-ray diffraction patterns of the ceramics sample of Series 2. As seen, in spite of the strong change in the Ga/Al ratio in the resulting samples compared with Series 1, all samples have a single garnet phase. Figure 5 shows X-ray diffraction patter in the vicinity of the major reflex. Inset shows calculated lattice parameters. Similarly to Series 1, reflections of the compounds obtained by mixing were measured in the interval between the reflection of the compounds, which were mixed. At the same time, the reflection of the GYAGG:Ce Ref5 sample practically coincides with the reflections obtained from the mixtures, with a median particle size below 4 μm.

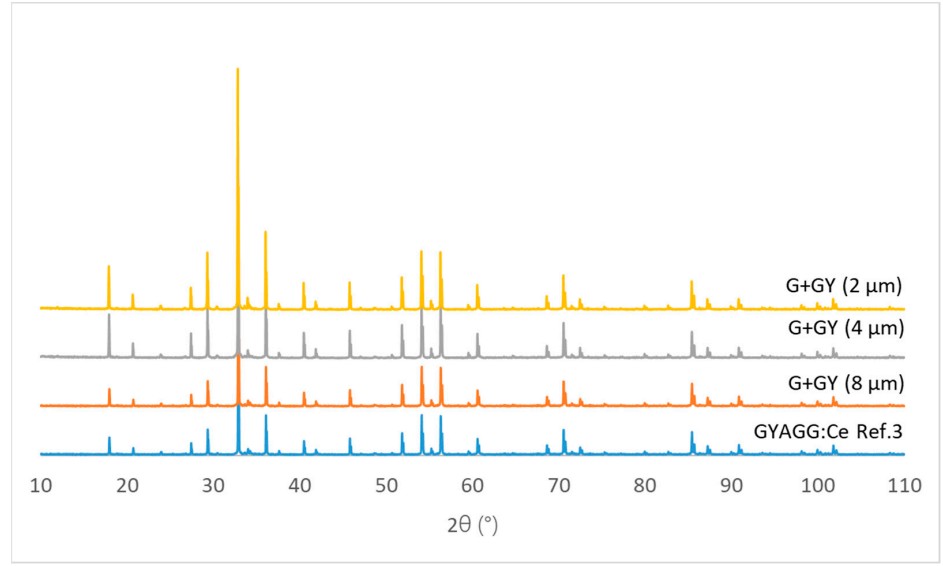

**Figure 2.** X-ray diffraction pattern of some samples of Series 1.

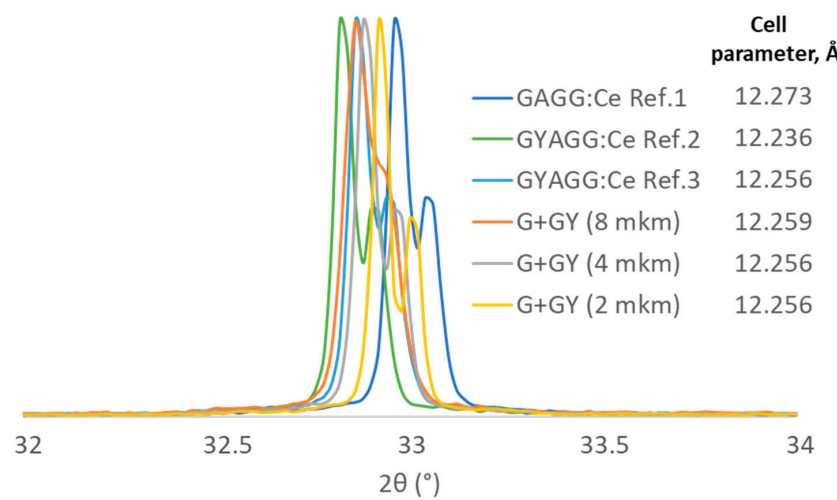

**Figure 3.** Details of the X-ray diffraction pattern in the vicinity of the major reflex in the interval 2θ 32 to 34 (°) obtained from the samples of Series 1. Intensity was normalized.

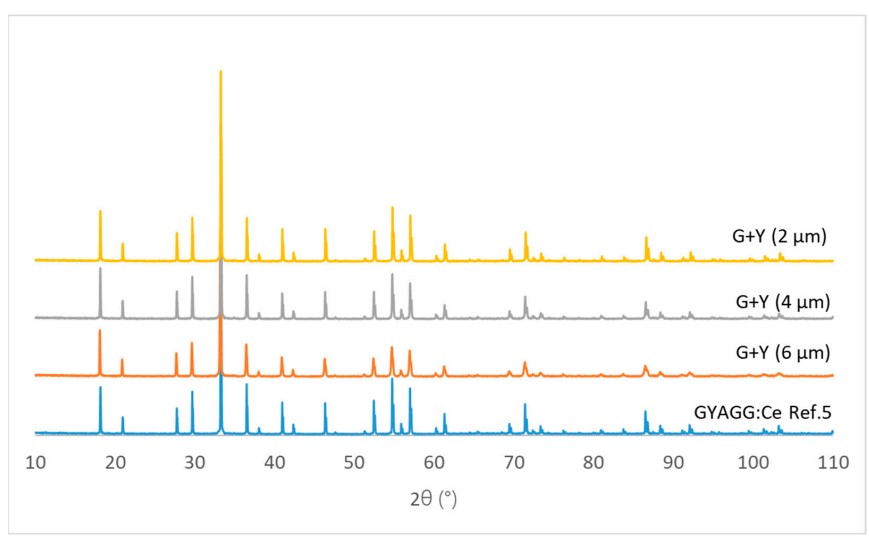

**Figure 4.** X-ray diffraction pattern of the samples from Series 2.

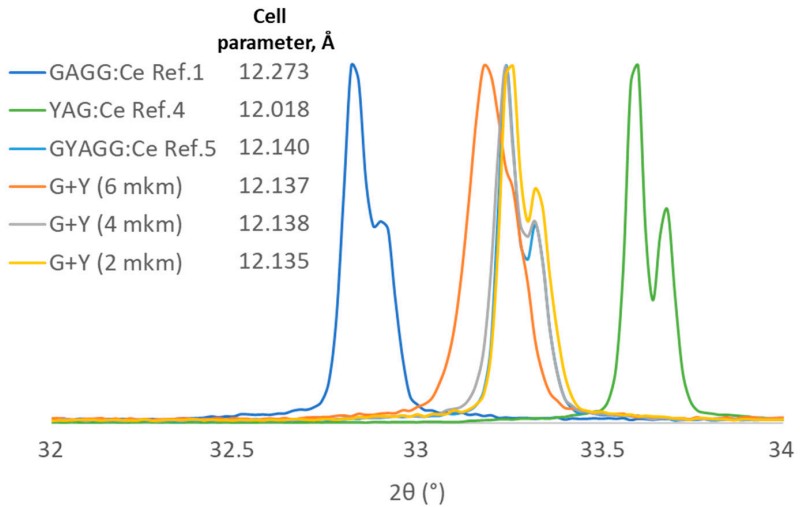

**Figure 5.** Details of the X-ray diffraction pattern in the major reflex range 2θ from 32 to 34 (°) from samples of Series 2. Intensity is normalized.

Figures 6 and 7 show the results obtained with the samples of Series 3. A remarkable difference between Series 2 and Series 3 is that no Gd was in the resulting compound. Reducing the factor of compositional disorder in the resulting compound does not affect its general property: all compounds had a pure garnet structure, and samples produced from the mixture were quite similar to that, which were obtained from the coprecipitated powder.

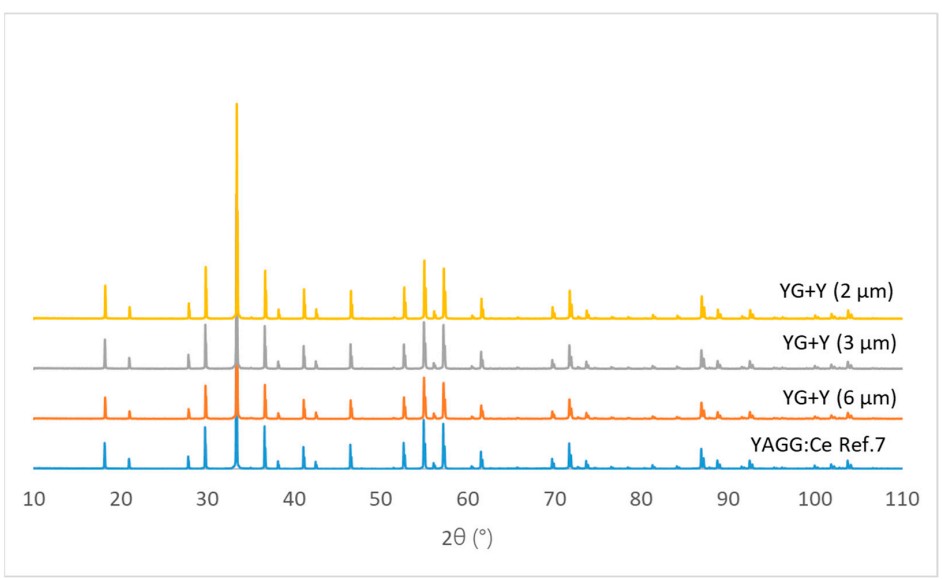

**Figure 6.** X-ray diffraction pattern of the samples from Series 3.

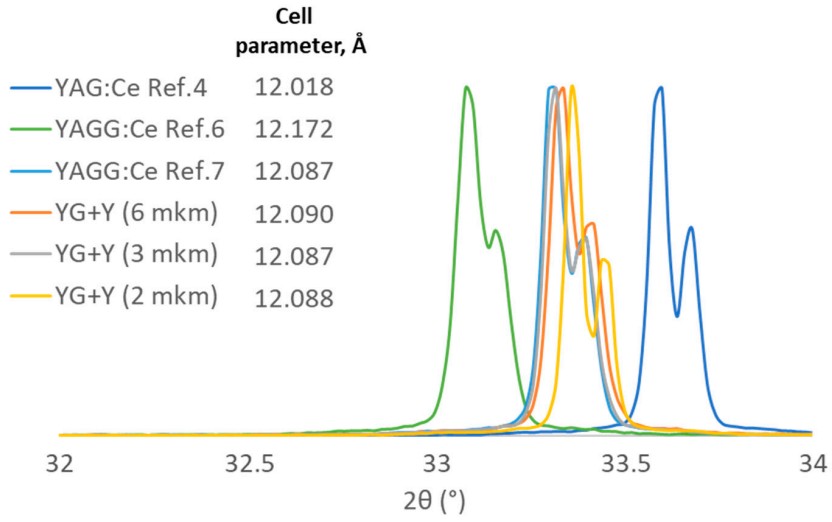

**Figure 7.** Details of the X-ray diffraction pattern in the major reflex in the interval 2θ from 32.5 to 34.5 (°) in samples from Series 3.

Figures 8–10 show the luminescence spectra of the samples from Series 1–3 upon excitation in the region of the first interconfigurational transition of $Ce^{3+}$ ions. The excitation was carried out both at the maximum of the band, and at wavelengths on the long-wavelength and short-wavelength wings of the band, corresponding to this transition. All series showed a similar result: the luminescence bands of the samples obtained from mixtures were located between the spectra of the mixture components. Moreover, the spectra coincide with those of the samples obtained by coprecipitation and do not depend on the granulometric composition of the mixed components.

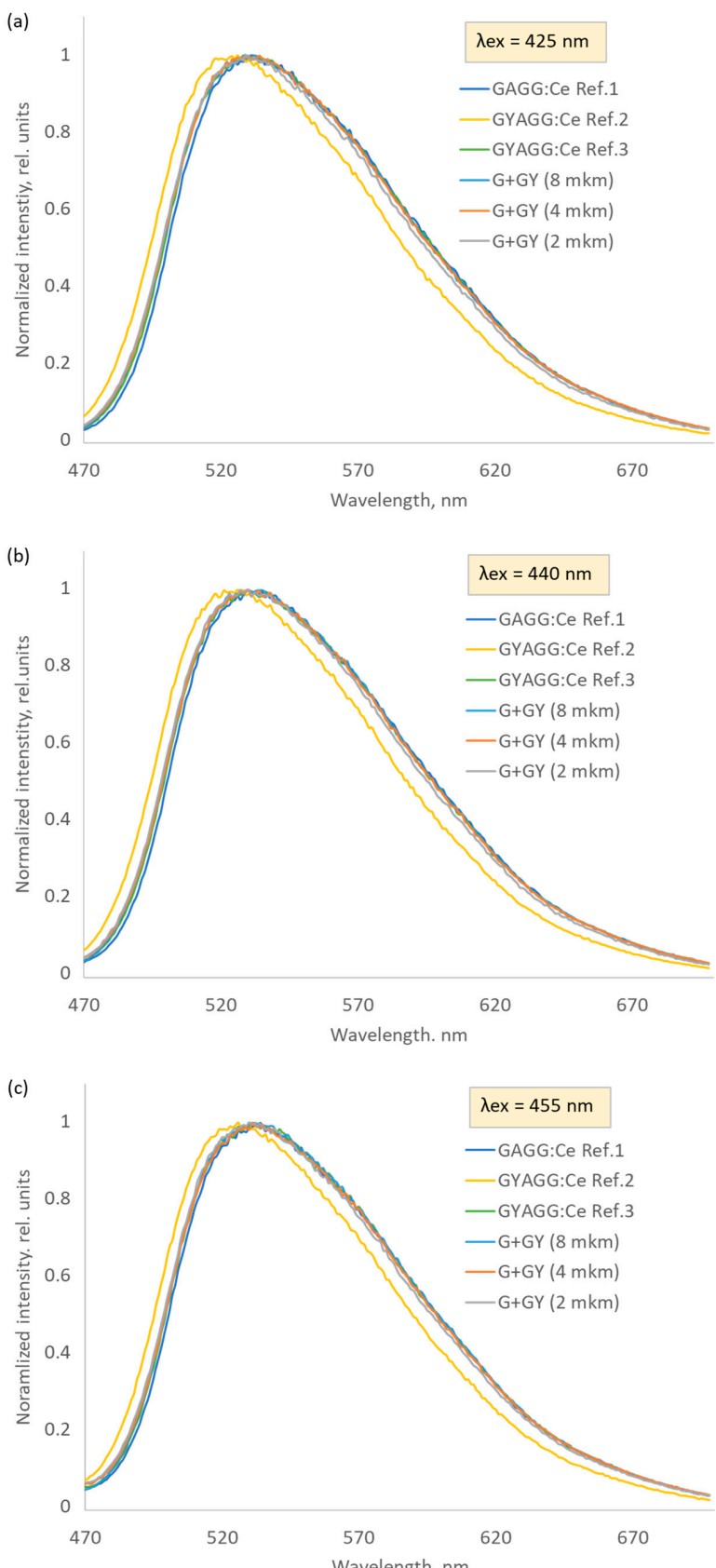

**Figure 8.** The room-temperature photoluminescence spectra of the samples in Series 1 at different excitations (indicated).

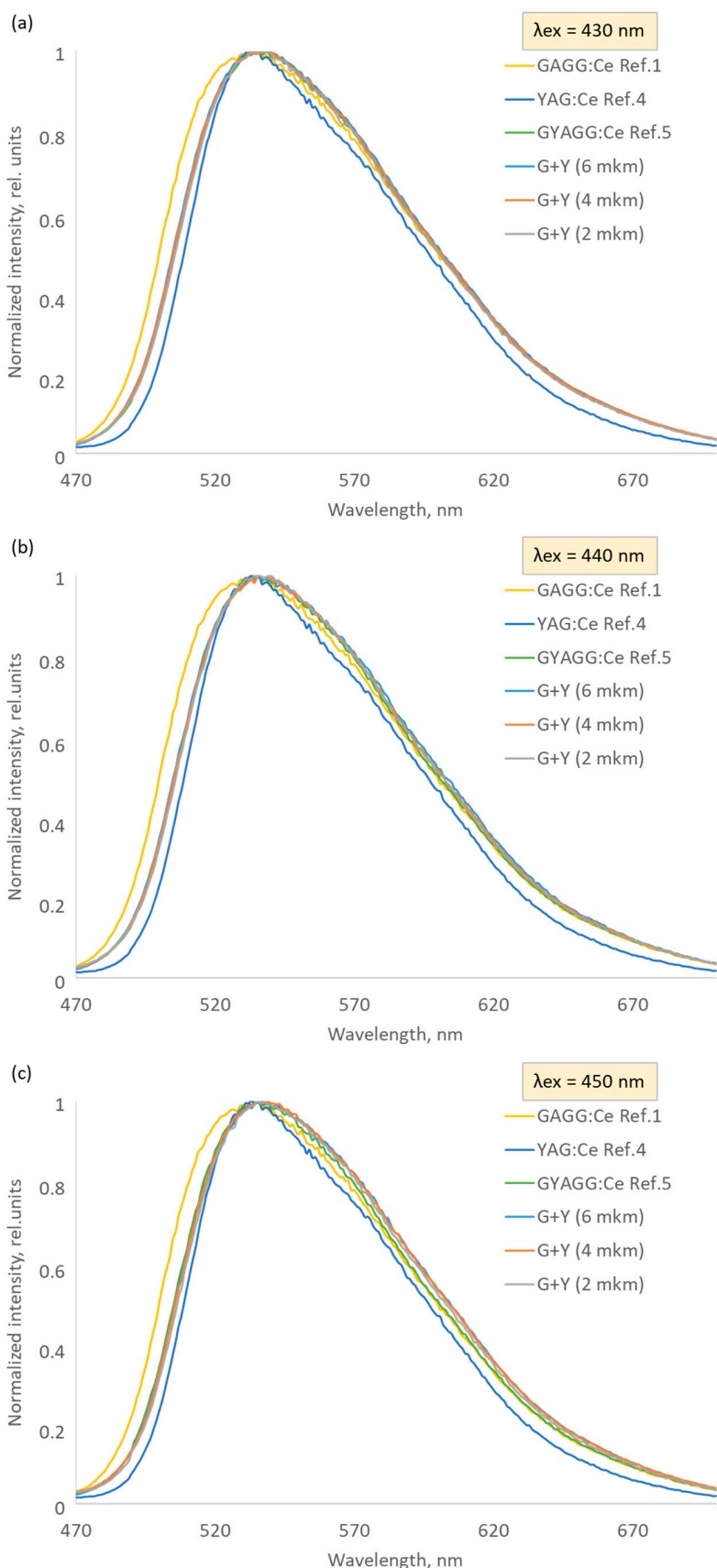

**Figure 9.** The room-temperature photoluminescence spectra of the samples in Series 2 at different excitations (indicated).

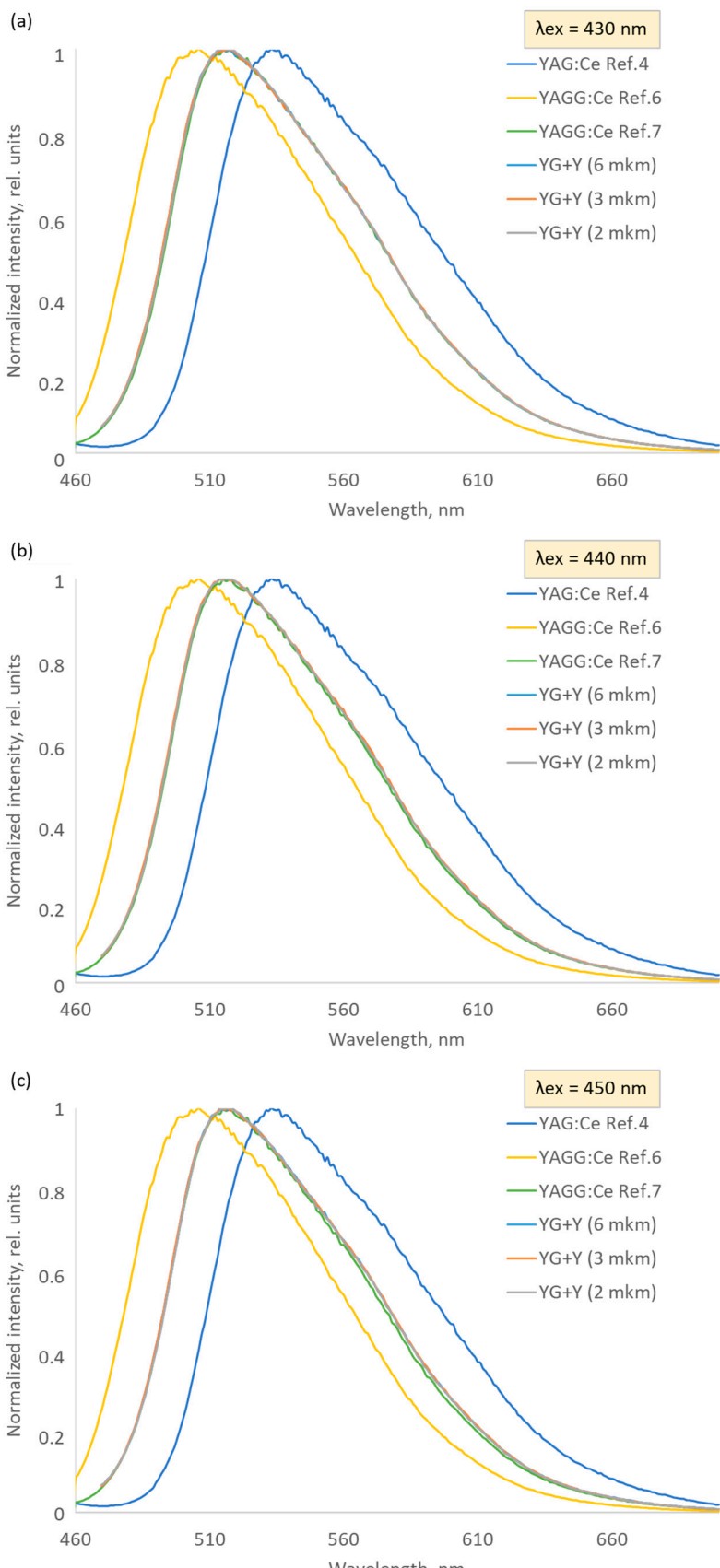

**Figure 10.** The room-temperature photoluminescence spectra of the samples in Series 3 at different excitations (indicated).

The scintillation kinetics of the samples of Series 1 and the representative samples of Series 2 and 3 are shown in Figures 11 and 12. Parameters of the scintillation kinetics and the relative LY of the samples are listed in Table 2.

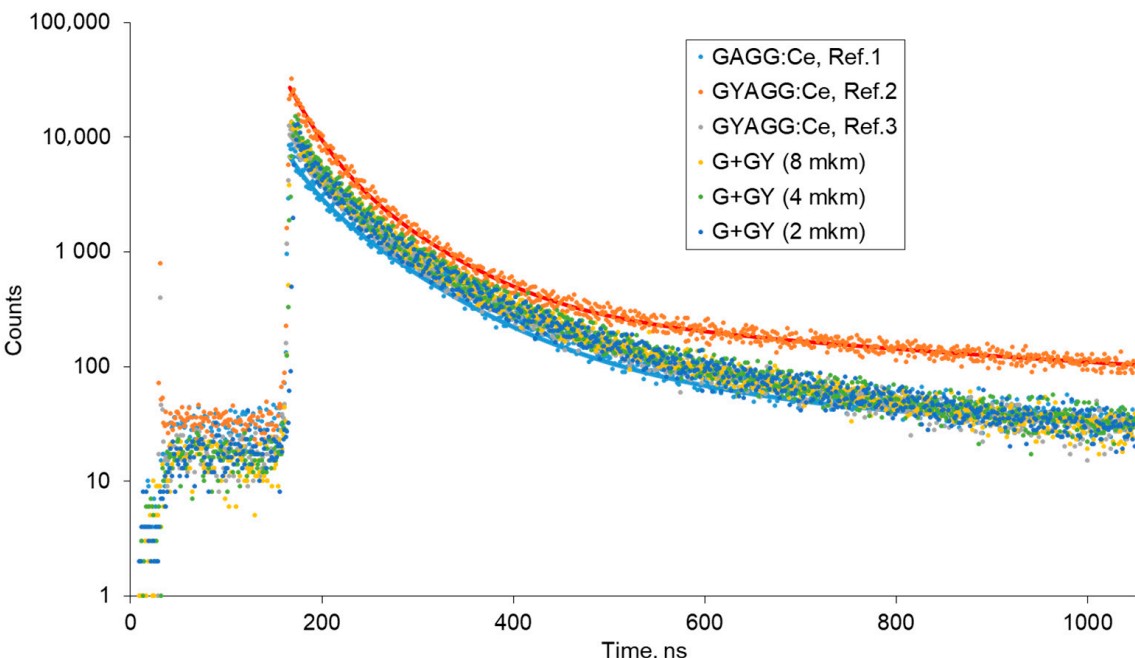

**Figure 11.** Room temperature scintillation kinetics of the representative samples of Series 1.

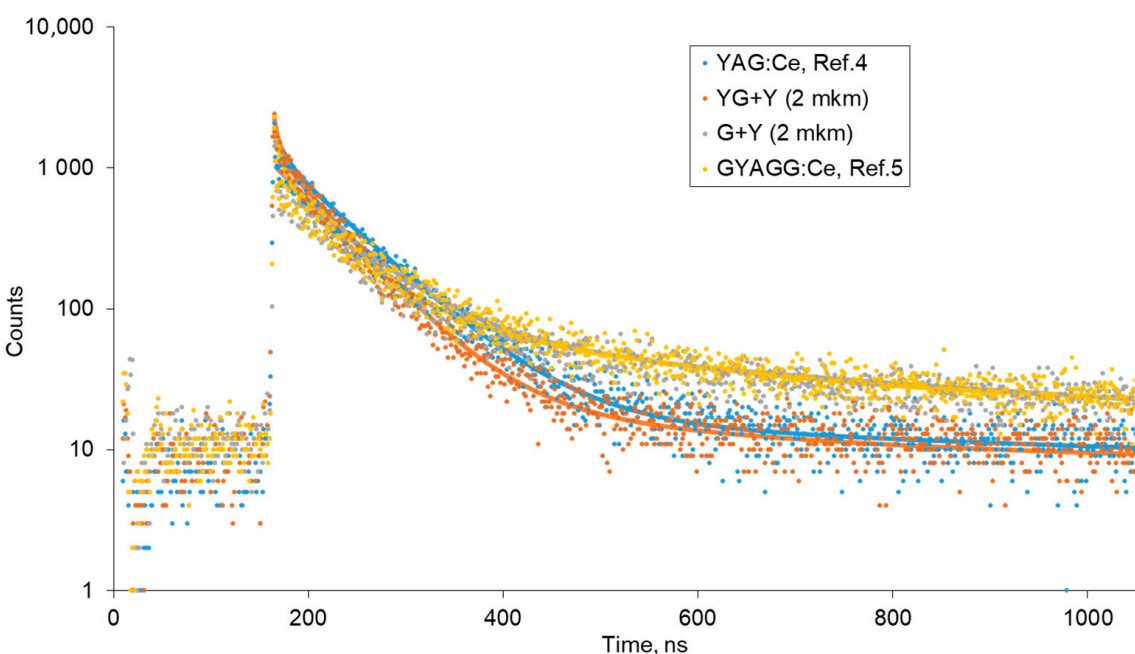

**Figure 12.** Room temperature scintillation kinetics of the representative samples of the Series 2 and 3.

**Table 2.** Relative light yield and parameters of the room temperature scintillation kinetics and of the representative samples of Series 1–3.

| Sample | Relative LY, rel. un. | Parameters of the Scintillation Kinetics, ns (%) |
|---|---|---|
| YAG:Ce single crystal with grinded surfaces | 1 | |
| | Series 1 | |
| GAGG:Ce Ref1 | 1.2 | 19 (16); 62 (63); 174 (21) |
| GYAGG:Ce Ref2 | 2 | 27 (44); 64 (42); 490 (14) |
| GYAGG:Ce Ref3 | 1.75 | 31 (46); 76 (39); 230 (15) |
| G+GY (8 μm) | 1.68 | 27 (37); 69(48); 249(15) |
| G+GY (4 μm) | 1.71 | 33 (52); 87 (38); 248 (10) |
| G+GY (2 μm) | 2 | 26 (43); 74 (43); 263 (14) |
| | Series 2 | |
| YAG:Ce Ref4 | 1 | 44 (19); 78 (73); 830 (8) |
| GYAGG Ref5 | 2 | 33 (22); 66 (42); 490 (35) |
| G+Y (6 μm) | 2 | |
| G+Y (4 μm) | 1.64 | |
| G+Y (2 μm) | 1.5 | 46 (26); 83 (37); 607 (36) |
| | Series 3 | |
| YAGG Ref6 | 1.4 | |
| YAGG Ref7 | 1.32 | |
| YG+Y (6 μm) | 1.17 | |
| YG+Y (3 μm) | 1.24 | |
| YG+Y (2 μm) | 1.2 | 21 (21); 58 (72); 330 (7) |

## 4. Discussion

It was recently demonstrated that annealing in oxygen can produce both translucent and transparent [29] ceramics of garnet-structure materials. When applied to materials containing gallium, this method of sintering was found to be promising, since it helped to solve the problem of gallium volatilization from the compound compact under heating. The close method of sintering in air was used to obtain ceramics from mixed powders. The obtained results show that crystal systems of the garnet structural type, in the form of ceramics with a higher level of disorder in the cationic subsystem, can be obtained by mixing compounds of the same structural type, but with a smaller number of cations in the composition. The approach used is based on a fairly wide region of phase homogeneity in the compounds of aluminum-gallium garnets. The cubic symmetry of the initial and resulting compounds provides the least energy-intensive mass transfer process without the formation of lower-symmetry phases, perovskite and monoclinic, at the formation of ceramics, as seen from the X-ray diffraction spectra. Note the close density of compounds obtained from mixtures with different granulometric compositions that were obtained in an identical production process. The data from Table 1 indicate that the calculated Rontgen density is quite close and does not depend on the average particle size of the mixed compounds. Of course, it should be noted that the intended particle size is the position of the maximum in the particle size distribution; the mixture contains particles of both smaller and larger sizes. Nonetheless, this does not impair its ability to produce high-quality ceramics, with the same structural type and spatial symmetry, that are uniform in volume. Thus, when using mixtures, the effective mass transfer and a uniform but random distribution of cations occur similarly to when utilizing coprecipitated material.

It has been proven that the luminescent properties of ceramics also do not depend on the granulometric composition of the mixture components used (Figures 8–10). A similarity of the spectra of the samples obtained by mixture also implies that, in a relatively short period of time, the case occurs when a disordered compound with a uniform and random distribution of cations in the crystal lattice forms. This also extends to impurity $Ce^{3+}$ ions and indicates that the crystallites of the compounds used for the mixture are completely diluted, and the composition of the compound is averaged. This is a rather remarkable fact, since the uniformization of the composition occurs without melting the crystalline

mass. Obviously, varying the weight fractions of the components selected for mixing makes it possible to control the position of the luminescence band maximum. Furthermore, in ceramic samples based on yttrium (Series 3) and lacking gadolinium, a sufficiently large tuning range of the luminescence band maximum ~40 nm can be obtained.

The scintillation properties of the obtained materials correlate with the data obtained by other authors for similar compositions [30–33]. However, we note some peculiarities. An increased concentration of Ce ions (x = 0.015) was used, which is at least five times higher than in single-crystal samples of close compositions. This causes the presence of a fast component in scintillations, of which the decay constant and the fraction in the kinetics depend on the series number of the samples. As already noted, the samples were prepared without codoping with magnesium ions. Therefore, the scintillation kinetics of all samples contained a slow component, the proportion of which varied by less than 20%. For Series 1, where the mixing of GAGG and GYAGG was used, there was good concurrence of both scintillation kinetics and LY at different granulometric compositions of the mixed powders. In series 2, when GAGG and YAG were mixed up, approximately the same trends were observed. In the latter series, even at a highly shifted Al/Ga ratio, the scintillation yield remains at the level of the samples when Al/Ga = 1 (Series 1). In Series 3, Gd ions were excluded from the crystal lattice, which lead to a decrease in the yield of scintillations. However, the general trend remains the same: both the yield of scintillations and the kinetics weakly correlate with the granulometric parameters of the powders used for mixtures.

## 5. Conclusions

The obtained experimental results convincingly show that crystalline ceramic compounds, of a garnet structural type characterized by having a high degree of compositional disorder due to a large number of cations in the lattice, can be obtained by combining powders of compounds of the same structural type, but with a smaller number of cations, under high-temperature annealing. The properties of such ceramics—structural, photoluminescent, and scintillation—weakly depend on the granulometric composition of mixed powders in the studied dimensional range and are close to those of ceramics obtained from powders synthesized by the co-precipitation method. The developed method is obviously suitable for crystalline systems that combine cubic symmetry with a relatively wide range of phase stability. Apparently, the developed method is quite universal for systems of cubic symmetry, which is confirmed by the study of high-luminosity ceramics obtained when mixing $Gd_3Al_2Ga_3O_{12}/(Gd,Y)_3Al_2Ga_3O_{12}$, $Y_3Al_5O_{12}/Gd_3Al_2Ga_3O_{12}$, and $Y_3Al_5O_{12}/Y_3Al_2Ga_3O_{12}$. We also concluded that for such compounds, after a relatively short period of sintering, the disordered compound with a uniform and random distribution of cations occurs in the crystal lattice. This also extends to impure $Ce^{3+}$ ions and indicates that the crystallites of the compounds used for the mixture are completely diluted, and the composition of the compound is averaged. We mark this effect as quite remarkable since the uniformization of the composition occurs without melting the crystalline mass.

**Author Contributions:** Conceptualization, M.K. and V.R.; methodology, V.S.; validation, D.T. and D.K.; investigation, A.B. and I.K.; writing—original draft preparation, D.K. and P.K.; writing—review and editing, V.P. and M.K. All authors have read and agreed to the published version of the manuscript.

**Funding:** This research received no external funding.

**Institutional Review Board Statement:** Not applicable.

**Informed Consent Statement:** Not applicable.

**Data Availability Statement:** No new and additional data are available.

**Acknowledgments:** The authors at NRC "Kurchatov Institute" acknowledge support from the Russian Ministry of Science and Education, Agreement No. 075-15-2021-1353. Analytical research was conducted using equipment of the «Research Chemical and Analytical Center NRC» «Kurchatov Institute» Shared Research Facilities under project's financial support by the Russian Federation, represented by The Ministry of Science and Higher Education of the Russian Federation, Agreement No. 075-15-2023-370 dd. 22.02.2023. Authors at Ural Federal University acknowledge partial support from the Ministry of Science and Education, project No. FEUZ-2023-0013 and program of strategic academic leadership "Priority 2030".

**Conflicts of Interest:** The authors declare no conflict of interest.

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
