# Peer review of "Advances of the Cubic Symmetry Crystalline Systems to Create Complex, Bright Luminescent Ceramics"

_photonics, doi:10.3390/photonics10050603_

Round 1

Reviewer 1 Report

In the current manuscript, a method to create compositionally disordered compounds having a high number of cations in the matrices, that utilizes the cubic spatial symmetry of the garnet-type crystalline systems is demonstrated. Further, the structural, luminescence and scintillation properties have been investigated for several mixed systems. Overall, the idea received my attention and the methodology is technically sound. However, there are some specific issues the authors should address by making modifications before we can proceed and positive action can be taken.

1. Be consistent in your colour code: if you have several figures in a publication highlight the same elements with the same colours in all of them (varying shades and intensity if necessary). This way the colour code helps your audience read the figures more efficiently.

2. The authors show optical properties in Figs. 8–10. I recommend the authors add a visible spectrum to the figure according to Fig. 5 of [Electronic and optical properties of heterostructures based on transition metal dichalcogenides and graphene-like zinc oxide. Sci. Rep. 2018, 8, 12009, doi:10.1038/s41598-018-30614-3] to make the story clearer and benefit the readers.

3. The decimal separators need to be consistent throughout the manuscript. The authors use point “.” in the main text while comma “,” in figures.

4. Use subfigures appropriately to condense the figure count. Subfigures should be closely related.

5. The authors investigated the optical properties of the systems. Have the authors noticed some papers documenting this point? i.e., [MoS2/ZnO van der Waals heterostructure as a high-efficiency water splitting photocatalyst: a first-principles study. Phys. Chem. Chem. Phys. 2018, 20, 13394–13399, doi:10.1039/C8CP00808F], [Strain effect on circularly polarized electroluminescence in transition metal dichalcogenides. Phys. Rev. Research 2020, 2, 033340, doi:10.1103/PhysRevResearch.2.033340] and the paper in point #2.

Author Response

Reviewer 1

The Reviewer comment

Author response

In the current manuscript, a method to create compositionally disordered compounds having a high number of cations in the matrices, that utilizes the cubic spatial symmetry of the garnet-type crystalline systems is demonstrated. Further, the structural, luminescence and scintillation properties have been investigated for several mixed systems. Overall, the idea received my attention and the methodology is technically sound. However, there are some specific issues the authors should address by making modifications before we can proceed and positive action can be taken.

1.Be consistent in your colour code: if you have several figures in a publication highlight the same elements with the same colours in all of them (varying shades and intensity if necessary). This way the colour code helps your audience read the figures more efficiently

Following the recommendation of the Reviewer, we updated the plots to reflect the particular color of each sample.

2.The authors show optical properties in Figs. 8–10. I recommend the authors add a visible spectrum to the figure according to Fig. 5 of [Electronic and optical properties of heterostructures based on transition metal dichalcogenides and graphene-like zinc oxide. Sci. Rep. 2018, 8, 12009, doi:10.1038/s41598-018-30614-3] to make the story clearer and benefit the readers.

The authors thank the Reviewer for interesting reference. It is incorporated in the part of the reference list connected with the introductory part.

We agree with the Reviewer that adding the spectrum to plots with luminescence curves may be useful. In practice, it is useful when wide-spectrum sources are considered. We describe relatively narrow bands, so to prevent overloading the figures, we prefer not to put strips of spectrum  in.

3.The decimal separators need to be consistent throughout the manuscript. The authors use point “.” in the main text while comma “,” in figures

Corrected

4.Use subfigures appropriately to condense the figure count. Subfigures should be closely related.

Corrected

5.The authors investigated the optical properties of the systems. Have the authors noticed some papers documenting this point? i.e., [MoS2/ZnO van der Waals heterostructure as a high-efficiency water splitting photocatalyst: a first-principles study. Phys. Chem. Chem. Phys. 2018, 20, 13394–13399, doi:10.1039/C8CP00808F], [Strain effect on circularly polarized electroluminescence in transition metal dichalcogenides. Phys. Rev. Research 2020, 2, 033340, doi:10.1103/PhysRevResearch.2.033340] and the paper in point #2.

The authors thank the Reviewer for interesting references. They are incorporated in the part of the reference list connected with the introductory part.

Reviewer 2 Report

Samples are referred to as Ref 1, 2 etc. This is meaningless to the reader, provide some indication of what the samples contain.

Rontgen density is not real

Scintillation kinetics is not real

Lots of wasted white space in figures

There are a lot of made-up terms in this paper, which is not acceptable

"Scintillation kinetic" curves need fitting to be meaningful

The English does not flow and needs a complete rewrite to not distract from the science.

Author Response

Reviewer 2

The Reviewer comment

Author response

Samples are referred to as Ref 1, 2 etc. This is meaningless to the reader, provide some indication of what the samples contain.

A special comment is provided to Table 1 at its bottom.

Rontgen density is not real

Corrected

Scintillation kinetics is not real

Scintillation kinetics approximation curves are added to the figures.

Lots of wasted white space in figures

Corrected where necessary.

There are a lot of made-up terms in this paper, which is not acceptable

Corrected where necessary.

"Scintillation kinetic" curves need fitting to be meaningful

Scintillation kinetics approximation curves are added to the figures.

Reviewer 3 Report

In this work, the authors synthesized several mixed systems, namely Gd3Al2Ga3O12/(Gd,Y)3Al2Ga3O12, Y3Al5O12/Gd3Al2Ga3O12 and, Y3Al5O12/Y3Al2Ga3O12, in order to give a method to create complex, bright luminescent ceramics. The research is interesting and systematically studies the luminescence behavior. Thus, I advise the publication after addressing the follow questions:

1.     Please supplement the energy level diagram for the luminescence mechanism analysis.

2.     The luminescence should be discussed with the local structure of the systems, and the crystal field strength should be mentioned for Ce3+.

3.     Why did author investigate this work? What is the novelty? Please give more clear introduction.

Minor editing of English language required。

Author Response

Reviewer 3

The Reviewer comment

Author response

In this work, the authors synthesized several mixed systems, namely Gd3Al2Ga3O12/(Gd,Y)3Al2Ga3O12, Y3Al5O12/Gd3Al2Ga3O12 and, Y3Al5O12/Y3Al2Ga3O12, in order to give a method to create complex, bright luminescent ceramics. The research is interesting and systematically studies the luminescence behavior. Thus, I advise the publication after addressing the follow questions:

1.Please supplement the energy level diagram for the luminescence mechanism analysis.

In this manuscript, we  describe  only Ce3+ activating ion. The energy levels diagram of the Ce ions is well established and discussed in [Auffray E., Augulis R., Fedorov A., Dosovitskiy G., Grigorjeva L., Gulbinas V., Koschan M., Lucchini M., Melcher C., Nargelas S., Tamulaitis G., Vaitkevičius A., Zolotarjovs A., Korzhik M. Excitation transfer engineering in Ce-doped oxide crystalline scintillators by codoping with alkali-earth ions // Physica Status Solidi (a). – 2018. – Vol. 215. – №. 7. – P. 1700798. DOI: https://doi.org/10.1002/pssa.201700798].

The authors agree with the Reviewer that the diagram will be important, in particular when a few activating centers are used.

2.The luminescence should be discussed with the local structure of the systems, and the crystal field strength should be mentioned for Ce3+.

The crystal field peculiarities for the compositionally disordered crystals are described in detail in the article by J. Ueda and S. Tanabe (doi.org/10.1016/j.omx.2019.100018). Differentially, we will quote this article in the publication dedicated to the more complex set of activating ions in the composition of the compound.

3.Why did author investigate this work? What is the novelty? Please give more clear introduction.

The following sentences in the introductory part of the article explain our motivation to seek new technological procedures to obtain highly compositionally disordered crystalline compounds.

“At the same time, an intention to move from binary and ternary compounds to more complex quaternary and quintuple compoistions, containing a larger number of cations, inevitably leads to problems with the precise control of the composition of the compound due to effects associated with the different solubilities of rare earth (RE) elements in the compound. A violation, for example, of the stoichiometry of the compound can lead to the leveling of the advantages introduced into the compound by compositional disorder in the cationic subsystem [21,22]”.

Reviewer 4 Report

The study titled as Advances of the cubic symmetry crystalline systems to create complex, bright luminescent ceramics is a well written manuscript. Most elements of the study are in accordance with the guidelines, and data are decently presented. However, throughout the manuscript, some syntax errors and a lack of clarity were observed. Therefore, I would recommend some minor suggestions before accepting the paper.

Comments to the author

1. The manuscript requires an overall English correction, and the introduction does not contain enough background to the study. Adding more literature to the introduction part would improve the quality of the manuscript considerably.

2.      SEM image of only one sample is provided in the manuscript (GAGG:Ce Ref1). To understand the morphology of the synthesized materials, need to add SEM image of all samples with an elaborated discussion on the structure.

3.      In page 2, line 79-80, some clarification is needed.

“Samples were not codoped with a small quantity of Mg to reduce a fraction of the slow component in the scintillation.”

Here, author need to clarify whether they doped Mg or not. If magnesium is not doped, then author must explain why Mg is mentioned here.

The manuscript requires an overall English correction. The language of the paper can be improved.

Author Response

Reviewer 4

The Reviewer comment

Author response

The study titled as Advances of the cubic symmetry crystalline systems to create complex, bright luminescent ceramics is a well written manuscript. Most elements of the study are in accordance with the guidelines, and data are decently presented. However, throughout the manuscript, some syntax errors and a lack of clarity were observed. Therefore, I would recommend some minor suggestions before accepting the paper.

1. The manuscript requires an overall English correction, and the introduction does not contain enough background to the study. Adding more literature to the introduction part would improve the quality of the manuscript considerably.

Following the recommendation of the Reviewer we complemented the introductory part of the article.

2.      SEM image of only one sample is provided in the manuscript (GAGG:Ce Ref1). To understand the morphology of the synthesized materials, need to add SEM image of all samples with an elaborated discussion on the structure.

Following to advice of the Reviewer we added SEM images of the representative samples to illustrate their similarity.

Figure, which will include 16 panels, overload the manuscript.

3.      In page 2, line 79-80, some clarification is needed.

“Samples were not codoped with a small quantity of Mg to reduce a fraction of the slow component in the scintillation.”

Here, author need to clarify whether they doped Mg or not. If magnesium is not doped, then author must explain why Mg is mentioned here.

An appropriate part of the manuscript is modified accordingly.

“Codoping of the GYAGG crystals with a small concentration of the magnesium Mg2+ (10-20 ppm) improves scintillation properties of the material, in particular the scintillation kinetics [23]. To distinguish effects of codoping from the effects that might be provided by applied technology, samples were not codoped with a small quantity of Mg to reduce a fraction of the slow component in the scintillation. Despite this, all samples did not show significant phosphorescence”.

Round 2

Reviewer 1 Report

The authors have responded to the queries put to them, and they have also incorporated the necessary changes needed. The whole presentation is clearer, with informative figures and complete references. In light of the above, I would recommend the publication of this manuscript.

Reviewer 3 Report

The authors have answered the reviewer questions adequately. I think that the manuscript can now be accepted.